# The Influence of Cadmium on Fountain Grass Performance Correlates Closely with Metabolite Profiles

**DOI:** 10.3390/plants12213713

**Published:** 2023-10-29

**Authors:** Zhaorong Mi, Pinlin Liu, Lin Du, Tao Han, Chao Wang, Xifeng Fan, Huichao Liu, Songlin He, Juying Wu

**Affiliations:** 1School of Horticulture and Landscape Architecture, Henan Institute of Science and Technology, Xinxiang 453003, China; mizr@hist.edu.cn (Z.M.);; 2Henan Provincial Engineering Research Center of Horticultural Plant Resource Utilization and Germplasm Enhancement, Xinxiang 453003, China; 3Institute of Grassland, Flowers and Ecology, Beijing Academy of Agriculture and Forestry Sciences, Beijing 100097, China

**Keywords:** cadmium stress, fountain grass, biological endpoint, amino acid, metabolic profiling, purine metabolism

## Abstract

The relationship between metabolite changes and biological endpoints in response to cadmium (Cd) stress remains unclear. Fountain grass has good Cd enrichment and tolerance abilities and is widely used in agriculture and landscaping. We analyzed the metabolic responses by detecting the metabolites through UPLC-MS and examined the relationships between metabolite changes and the characteristics of morphology and physiology to different Cd stress in fountain grass. Our results showed that under Cd stress, 102 differential metabolites in roots and 48 differential metabolites in leaves were detected, with 20 shared metabolites. Under Cd stress, most of the carbohydrates in leaves and roots decreased, which contributed to the lowered leaf/root length and fresh weight. In comparison, most of the differential amino acids and lipids decreased in the leaves but increased in the roots. Almost all the differential amino acids in the roots were negatively correlated with root length and root fresh weight, while they were positively correlated with malondialdehyde content. However, most of the differential amino acids in the leaves were positively correlated with leaf length and leaf fresh weight but negatively correlated with malondialdehyde content. Metabolic pathway analysis showed that Cd significantly affects seven and eight metabolic pathways in the leaves and roots, respectively, with only purine metabolism co-existing in the roots and leaves. Our study is the first statement on metabolic responses to Cd stress and the relationships between differential metabolites and biological endpoints in fountain grass. The coordination between various metabolic pathways in fountain grass enables plants to adapt to Cd stress. This study provides a comprehensive framework by explaining the metabolic plasticity and Cd tolerance mechanisms of plants.

## 1. Introduction

The development of modern agriculture and industry has given rise to serious heavy metal pollution in the environment, of which cadmium (Cd) has become a global concern [1]. Cd, one of the most toxic heavy metals, disturbs the regular physiological metabolism of plants and animals, threatening human health by delivering it into the food web [2]. Cd stress can decrease stomatal density [3], inhibit photosynthesis, transpiration, biomass accumulation, and decreased water use efficiency [4], resulting in root necrosis and leaf chlorosis [5]. These morphological and physiological changes under Cd treatment are all caused by variations in metabolites and metabolic pathways; however, the changes in metabolites and metabolic pathways under Cd treatment are highly uncertain [6]. Metabolomics can effectively identify specific metabolites as biomarkers of specific stress responses, thereby gaining insight into the underlying metabolic mechanisms that counter stress to reveal cellular activities and organism responses to environmental alterations [7]. Previous studies have shown that the type and relative content of primary metabolites such as amino acids, organic acids, sugars, and sugar alcohols are modulated, as these are important for heavy metal sequestration, detoxification, and tolerance [8]. Additionally, the contents and varieties of secondary metabolites changed under Cd stress [9]. For example, Cd stress increased the amino acid contents in the leaves while decreasing the amino acid contents and the sugar contents in the roots [10]. However, the metabolite profile in the leaves of *Catharanthus roseus* showed that cadmium tolerance could be related to the differential accumulation of secondary metabolites [11]. In addition to the metabolite changes, the alterations in the number and types of metabolic pathways under Cd also varied with the intensity of the cadmium treatment, the time of cadmium treatment, and plant tissue. Cadmium influenced seven metabolic pathways in the roots, one in the stems, and three in the leaves of pumpkins [4]. In the roots of *Brassica juncea*, eight metabolic pathways were significantly changed at 48 h after Cd treatment, and four metabolic pathways were significantly changed on the seventh day after Cd treatment, respectively [6]. Another study found that there were thirteen metabolic pathways in relation to the differential metabolites in the roots of *Sedum plumbizincicola* under low Cd treatment and six metabolic pathways related to the differential metabolites in the roots of *S. plumbizincicola* under high Cd treatment [12]. The alterations in metabolites and metabolic pathways under Cd treatment could lead to changes in the plant’s internal and external biological endpoints (BEs) [13]. However, few studies have explored the relationships between changes in metabolites and BEs. For example, orthogonal partial least squares (OPLS) analyses were applied to explore the relationships between metabolite changes and fresh weight in pumpkin tissues [4].

The monocots included about 70,000 species, which occupied about a quarter of all angiosperms. About 12,000 species of monocots belong to the true grasses (Poaceae), which are economically the most important family of monocots. Fountain grass is a fast-growing, low-cost, high-yielding, and adaptable monocot plant with attractive leaves and spikes in summer, autumn, and winter [14]. Because of these advantages, there are many fountain grass species that are spectacularly and widely used as ornamental grass in landscaping [15]. Fountain grass is also an excellent forage widely cultivated in tropical and subtropical areas and good for ecological restoration [16]. Previous studies have explored the Cd enrichment ability of fountain grass cultivars [17] and evaluated changes in the structural and physiological leaf traits of fountain grass under Cd stress [3]. To the best of our knowledge, the changes in metabolic pathways and the relationship between the changes in metabolites and BEs of fountain grass is still unknown. Filling this knowledge gap could facilitate the control of urban cadmium pollution and the understanding of the adaptation mechanisms of monocots to cadmium stress.

In this study, fountain grass was treated under different Cd stress levels to (1) assess the potential effects of Cd stress on the growth and physiology of fountain grass, (2) provide a comprehensive mechanism of Cd stress combined with metabolic profile and BEs [such as leaf/root length, leaf/root fresh weight, and leaf/root malondialdehyde (MDA) content], and (3) illustrate the metabolic pathway linking the metabolite changes and biologic endpoints under Cd stress.

## 2. Results and Discussion

### 2.1. Cadmium Accumulated More in Roots Than in Leaves

As shown in Table 1, the Cd content was not detected in the CK in leaves and roots. The Cd content of the leaves under Cd_10_ was 1.75 times that of Cd_5_, and the Cd content of the roots under Cd_10_ was 2.10 times that of Cd_5_. Similar to our results, Cd accumulation in the roots and leaves increased gradually with the increase in the Cd content in the environment [4]. In the Cd_5_ treatment, the Cd content of the roots is 12.50 times that of the leaves, and the transfer factor is 8.34%. In comparison, the Cd content of the roots was 15.00 times that of the leaves, and the transfer factor was 6.77% under the Cd_10_ treatment. Previous studies have also found that Cd accumulated more in the roots than in the leaves of two fountain grass cultivars at certain Cd stress levels [17]. In addition, the transfer factor in our study was slightly lower than in the study of Chen et al. (2020); however, this was still less than one, indicating the Cd ions preferentially piled up in the roots [18]. When Cd entered the plant, it first accumulated in the root tip before entering the xylem through the symplastic pathway or apoplastic pathway. Then, it was transported into the vessel by the transporters on the tonoplast and plasma membrane and finally transferred to the leaves with plant transpiration [19]. Other studies have also shown that Cd ions mostly accumulate in the roots [20], which could be attributed to the barrier effect of roots on the transportation of Cd to the leaves [19].

### 2.2. The Negative Effects of Cd Stress on Morphological Characteristics

The leaf length significantly decreased by 22.10% and 47.10% under Cd_5_ and Cd_10_, respectively. Similar to the response of leaves, the root length also significantly decreased by 16.04% and 72.50% under Cd_5_ and Cd_10_, respectively (Figure 1A and Appendix A), illustrating how Cd stress significantly limited the elongation of the leaves and roots. Correspondingly, the leaf fresh weight, root fresh weight, and total fresh weight significantly decreased by 20.59%, 69.13%, and 35.74%, respectively, under Cd_10_ (Figure 1B,C). The toxicity of Cd could reduce the uptake and translocation of water and nutrients, increasing oxidative damage, disrupting the metabolism, and finally handicapping the mass accumulation of leaves and roots [21]. Some studies have also reported that leaf length and leaf fresh weight significantly decreased under Cd stress [4]. Due to a larger decrease in the root fresh weight compared with the leaf fresh weight, the root/shoot ratio significantly decreased by 61.00% under Cd_10_ (Figure 1D), indicating that roots were subjected to more toxic effects than leaves. The root/shoot ratio decreased with the increment in the Cd concentration (Figure 1D), demonstrating that the toxic effects of Cd on plants were concentration-dependent. The higher the Cd concentration, the more toxicity the Cd exerts on the tissues. Due to the particularly higher accumulation of Cd in the roots than in the leaves, the root morphological characteristics showed a larger decrease compared with the leaves.

### 2.3. MDA Content, Soluble Protein Content, and SOD, POD, and CAT Activity Changes under Cd Stress

With the increase in Cd concentration, the contents of MDA and soluble proteins in the leaves and roots tended to increase. Among these, the MDA content of leaves under Cd_5_ and Cd_10_ significantly increased by 17.25% and 65.88%, respectively. In comparison, the MDA content of roots under Cd_5_ and Cd_10_ increased by 33.17% and 81.91%, respectively (Figure 2A). MDA is a highly reactive tri-carbon dialdehyde that is related to the level of plant growth inhibition and is recognized as a biomarker of oxidative stress. Our results are in parallel with the study of Leng et al. (2021), who stated that Cd stress elevated the content of MDA both in the roots and leaves, closely relating to linolenic acid metabolism and tryptophan metabolism [22]. The MDA content of the leaf/root increased with the increment in Cd stress, indicating that Cd-induced lipid peroxidation was exacerbated as Cd toxicity was aggravated.

Similarly, the soluble protein content of leaves and roots significantly increased by 13.35% and 31.07% under Cd_10_, respectively (Figure 2B). Soluble proteins are significant components of numerous plant enzymes, reflecting the comprehensive metabolism of plants [23]. Soluble proteins play a critical role in increasing the amount of functional protein needed to maintain normal metabolic activities and improve the plant’s resistance to environmental stress [24]. According to a proteomics analysis of wheat under Cd stress, the number of upregulated differential expression proteins was higher than the downregulated [25]. Similar to previous studies, the content of soluble proteins in the leaves of a hybrid *Pennisetum* increased with Cd stress aggravation [26]. Under Cd stress, plants could synthesize some unique Cd-induced proteins, which occupy more than 10% of the total proteins [27]. These could be the reasons that the soluble protein content increased under Cd stress.

The SOD activity of roots under Cd_5_ and Cd_10_ significantly decreased by 9.31% and 17.47%, respectively (Figure 2C). The POD activity of leaves and roots under Cd_10_ significantly increased by 28.61% and 53.07%, respectively (Figure 2D). The CAT activity of the leaves and roots under Cd_10_ also increased by 88.98% and 151.61%, respectively (Figure 2E). With the increase in Cd concentration, the activities of SOD in the leaves and roots showed a decreasing trend, while the activities of POD and CAT showed an increasing trend. SOD dismutases O_2_¯ molecules to H_2_O_2_ and hence can decrease OH¯ formation through the metal-catalyzed Haber-Weiss reaction [28]. POD is one of the key enzymes, together with CAT, that participates in the breakdown and decomposition of H_2_O_2_ [29]. Similar to our results, the SOD activities of the leaves showed little change under a low Cd concentration; however, the SOD activities of the roots decreased under a high Cd concentration [30]. The dominant antioxidant enzymes in different plants are different. Similar to our results, Xu et al. (2012) also found that under cadmium stress, the POD and CAT activities of the leaves and roots of *Solanum torvum* increased significantly, while SOD activity remained basically unchanged [31]. The Cd content in the leaves was much lower than that in the roots; therefore, SOD activities in leaves were not significantly affected under Cd stress. However, the SOD activities in roots decreased under Cd treatment. Due to the higher Cd content in roots, the increased range of MDA, soluble protein, POD, and CAT in the roots was larger than in leaves under the same Cd concentrations.

### 2.4. Metabolite Changes under Cd Stress and Their Relationships with BEs

Cd treatment not only altered the BEs (e.g., leaf/root length, leaf/root fresh weight, MDA content), but also changed the content of different metabolites both in the leaves and roots of fountain grass. There were 48 metabolites detected as differential metabolites in the leaves (Figure 3A, Appendix A), and 102 metabolites in the roots (Figure 3B, Appendix A) were detected as differential metabolites, with 20 shared in both the leaves and roots. Then, these differential metabolites were clustered based on their functions/pathways into several categories, such as phytohormones, carbohydrates, amino acids, organic acids, lipids, nucleotides, benzenoids, phenylpropanoids, and polyketides. As shown in Figure 3, the metabolites of CK and Cd stress were classified into two categories by hierarchical clustering.

#### 2.4.1. The Changes in Differential Amino Acids under Cd Stress

Under Cd stress, 23 differential amino acids increased in the roots. It has been found that the carboxylate (-COO) and amine (-NH_2_) groups of amino acids can form complexes primarily with heavy metal ions, intensifying the heavy metal tolerance of plants [32,33]. Among these 23 differential amino acids, the content of arginine (FC = 3.28), aspartate (FC = 3.23), glutamate (FC = 3.36), tryptophan (FC = 4.14), alanine (FC = 2.55), and valine (FC = 2.05) in roots increased under Cd stress. Previous studies have also found that the arginine, aspartate, and glutamate content in the roots of *B. juncea* [6], the levels of isoleucine and valine in the roots of *Solanum nigrum* [34], and the content of tryptophan in the roots of *Noccaea caerulescens* increased under Cd stress [35]. The accumulation of tryptophan is beneficial to prevent redox damage to peptides and proteins [36]. Tryptophan can be dioxygenated by cracking the pyrrole ring to produce N-formylkynureine, which was found to be associated with 17 proteins involved in redox reactions. Therefore, the increase in tryptophan could contribute to the accumulation of soluble proteins [37]. In spite of the increase in proteinogenic amino acids, the content of most of these non-proteinogenic amino acids also increased under Cd stress in the roots, such as γ-aminobutyric acid (GABA, FC = 2.51), O-acetyl-l-serine (FC = 2.78) and pipecolic acid (FC = 3.91). GABA, O-acetyl-l-serine, and pipecolic acid have been found to increase as a result of their exposure to heavy metal stress, which acts to enhance oxidative stress resistance or as an osmo-protectant [38,39,40]. Interestingly, the content of 4-guanidinobutyric acid (FC = 4.81), Gly-Leu (FC = 2.39), ornithine (FC = 2.17) increased, but the content of asparagine (FC = 0.5), pantethine (FC = 0.49), s-adenosyl-l-homocysteine (FC = 0.46), 3-methyl-l-histidine (FC = 0.44), cysteine-s-sulfate (FC = 0.4), histidine (FC = 0.38), n-acetylhistidine (FC = 0.37), acetyl-dl-leucine (FC = 0.36), and saccharopine (FC = 0.22) decreased in leaves under Cd stress. Similar to our findings, the asparagine in plants under Cd stress was significantly lower than in the control [41]. The content of histidine in leaves is lower than that in the control for *N. caerulescens* under Cd treatment [35]. There were six shared differential amino acids in the leaves and roots, among which five showed similar change trends both in the leaves and roots. Similar findings have been reported in the differential amino acids shared in the roots and leaves of sunflowers, where the content of some amino acids was elevated, such as 4-aminobutyric acid and betaine; however, the content of some other amino acids was reduced, such as glutamine and pyroglutamic acid [42]. Parallel to our findings, the levels of ornithine in the roots and leaves of *Panicum maximum* cv. Massai increased under Cd stress [43]. Ornithine mainly acts on signal pathways controlling the synthesis of proline, arginine, putrescine, and GABA, conferring the importance of this amino acid to Cd hyperaccumulations [43].

#### 2.4.2. The Variations in Differential Carbohydrates under Cd Stress

There were seven differential carbohydrates in the roots, among which the content of n-acetyl-d-glucosamine 6-phosphate (FC = 0.32), 2-deoxyribose 5-phosphate (FC = 0.31), 2′-deoxy-d-ribose (FC = 0.29), 1-deoxy-d-xylulose 5-phosphate (FC = 0.25), and UDP-glucose (FC = 0.22) decreased while ribitol (FC = 2.5) and galactinol (FC = 2.4) increased. Additionally, there were eight differential carbohydrates in the leaves, among which the content of gentianose (FC = 0.45), 1-deoxy-d-xylulose 5-phosphate (FC = 0.44), UDP-d-glucose (FC = 0.43), perseitol (FC = 0.4), and n-acetyl-d-glucosamine 6-phosphate (FC = 0.27) decreased while the content of threitol (FC = 3.59), galacturonic acid (FC = 2.58), and maltotriose (FC = 2.22) increased. Galactinol can scavenge ·OH as efficiently as glutathione to abate oxidative stress under Cd treatment [44], while ribitol serves as a protectant against many abiotic stresses, such as heavy metal stress [45] and cold stress [46]. It has been reported that maltotriose could be an indicator of carbon pool reallocation under abiotic stress, and the increase of maltotriose could indicate starch hydrolysis countering Cd stress [47]. In line with the study of Sun et al. (2020), threitol also accumulated in the leaves under Cd stress [44]. It should be noted that changes in metabolites are associated with cell wall construction. Galacturonic acid is an important component of pectin, which approximately accounts for 70% of the cell [48]. The methyl-esterification of galacturonic acid determines the negative charge in pectin and thus determines its metal binding quantity [49]. Therefore, the increase in galacturonic acid in the leaves might enhance the content of Cd bound by the cell wall. In addition, Cd stress also reduced the UDP-glucose content, which is a shared differential product of roots and leaves. UDP-glucose, the activated form of glucose, serves as a precursor for the synthesis of UDP-glucuronic acid. The conversion of UDP-glucose to UDP-glucuronic acid is the first step of nucleotide sugar synthesis (UDP-apiose, UDP-arabinose, and UDP-xylose) [50]. The UDP-glucose precursors were utilized by cellulose synthase to elongate the cellulose chain, which is the main component of the cell wall [51]. Therefore, the decrease in UDP-glucose could limit the elongation of leaves and roots. Compared with CK, UDP-glucose decreased by 78.09% and 56.50% in the roots and leaves, respectively, indicating the greater inhibition effect of Cd ions on roots compared with leaves.

#### 2.4.3. Alterations in the Differential Lipids Induced by Cd Stress

Our study found that the content of 5-HETE (FC = 4.16), stearidonic acid (FC = 3.68), 4-oxoretinol (FC = 3.1), mesaconic acid (FC = 2.62), traumatic acid (FC = 2.25), and 3-hydroxy-3-methylglutaric acid (FC = 2.17) increased and only linoleic acid (FC = 0.09) decreased in the roots. 5-HETE was also found to accumulate concomitantly with an oxidative burst in the roots under Cd stress [52]. Parallel to our findings, stearidonic acid and 3-hydroxy-3-methylglutaric acid were found to be increased under Cd stress [44,53]. According to Wang et al. (2021), traumatic acid in the roots was found to increase under stress [54]. In line with our results, linoleic acid has been reported to be downregulated in *S. plumbizincicola* under Cd stress [12]. The content of 1-palmitoyl-sn-glycero-3-phosphocholine (FC = 3.55), progesterone (FC = 2.4), and cucurbitacin d (FC = 2.23) increased, and the content of 11-beta-hydroxyandrosterone (FC = 0.45), trans-traumatic acid (FC = 0.43), oxandrolone (FC = 0.43), 2-isopropylmalic acid (FC = 0.4), endo-borneol (FC = 0.31), and mesaconic acid (FC = 0.29) decreased in the leaves under Cd stress. The increase in abscisic acid could lead to a significant decrease in trans-traumatic acid content under Cd stress [55,56]. 2-isopropylmalic acid is the precursor of malic acid [57]; therefore, a decrease in 2-isopropylmalic acid could lead to a decrease in malic acid. By contrast, 1-palmitoyl-sn-glycero-3-phosphocholine and progesterone increased in the leaves. Progesterone, which is naturally present in plants, could significantly reduce the negative effects of adverse environments, stimulate antioxidant enzyme activity, reduce cell membrane permeability, and improve the efficiency of PSII [58]. Mesaconic acid was a differential metabolite shared in both the roots and leaves in our results. Similar to our results, mesaconic acid in the leaves was significantly decreased with late exposure to soil imidacloprid [59].

#### 2.4.4. Most of the Differential Nucleotides in Roots Decreased under Cd Stress

We found nineteen differential nucleotides and derivatives in the roots under Cd treatment, of which twelve decreased and seven increased (Appendix A). In comparison, there were five differential nucleotides in the leaves, of which only guanosine (FC = 0.36) decreased while guanosine 5′-monophosphate (FC = 3.28), adenosine monophosphate (FC = 2.26), auanosine diphosphate (FC = 2.09), and uridine 5′-monophosphate (FC = 2.02) increased. Previous studies reported that the content of cytidine 5′-diphosphate decreased under heavy metal stress [5]. Under stress conditions, plants inhibited the excessive accumulation of reactive oxygen species by consuming excessive reduction power in plant cells, thereby decreasing the NADH content [60]. Similar to our studies, the content of flavine mononucleotide and 1-methyladenosine in the roots decreased [61,62]. The increase in adenosine monophosphate content could promote Cd transport to vacuoles and lower Cd toxicity [63].

#### 2.4.5. The Differential Organic Acid Variations under Cd Treatment

Under Cd treatment, there were four differential organic acids in the roots, among which the content of glyoxylate (FC = 3.22) and glyceric acid (FC = 2.21) increased, while the content of 3-Hydroxybutyric acid (FC = 0.48) and gyruvaldehyde (FC = 0.26) decreased. In line with our results, glyceric acid in the roots of cucumber increased under Cd stress [44]. Under Cd treatment, there were four different organic acids in leaves, among which only oxalate (FC = 3.17) increased while the content of citrate (FC = 0.44), malic acid (FC = 0.43), and glutaric acid (FC = 0.37) decreased. The increase of oxalate in the leaves has been reported to be beneficial for plants to counter Cd toxicity [64]. In line with our results, the glutaric acid content in the leaves of cucumber decreased under Cd stress [44].

#### 2.4.6. The Differential Phytohormone Changes Induced by Cd Ions

Our study found four differential phytohormones in the roots under Cd stress, among which the content of indole-3-butyric acid (FC = 5.79) and phenylacetic acid (FC = 5.08) increased under Cd stress, while the content of melatonin (FC = 0.45) and kinetin (FC = 0.21) decreased. Previous research found that indole-3-butyric acid could reduce Cd toxicity in the barley roots [65], and the increase in indole-3-butyric acid might lead to an increase in phenylacetic acid [66]. It has been found that exogenous kinetin could increase the fresh and dry weights of roots and shoots of sorghum plants under Cd stress [67]. Therefore, the Cd-induced decrease in kinetin could lead to a decrease in root length and root fresh weight. Indole-3-acetamide (IAM) is a precursor of indoleacetic acid (IAA), which is a well-known growth regulator. The lower IAM content could lead to a decrease in IAA, therefore handicapping plant growth under Cd stress [68].

#### 2.4.7. The Alterations of Benzenoids, Phenylpropanoids, and Polyketides under Cd Stress

We found the content of hydroquinone (FC = 3.97 in leaves and FC = 3.76 in roots) shared both in the leaves and roots was enhanced under Cd stress, which is in line with the study of Borgo et al. (2022) [69]. Previous studies showed that exogenous hydroquinone inhibited cell growth and enhanced apoptosis gene expression [70]. Among the differential phenylpropanoids and polyketides, the bisdemethoxycurcumin in the roots showed a sharp increase (FC = 23.93) under Cd stress, which could effectively reduce oxidative stress [71]. Neohesperidin also increased under Cd stress, indicating a higher heavy metal stress tolerance [72]. Our results showed that genistein (FC = 4.83 in leaves and FC = 6.12 in roots), p-coumaric acid (FC = 3.00 in leaves and FC = 8.76 in roots), and 3-hydroxy-4-methoxycinnamic acid (FC = 2.01 in leaves and FC = 9.31 in roots) were differential metabolites that were shared in both leaves and roots, which all increased more in the roots than in the leaves under Cd stress. Similar to Yadav et al.’s (2021) research, genistein has been found to increase under As stress in the roots, young leaves, and old leaves of rice [62]. The function of some metabolites has been relatively clear; however, the functions of a lot of plant products remain unknown due to the large diversity and quantity of plant metabolites.

### 2.5. The Relationships of Main Metabolite Changes with BEs under Cd Stress

In the OPLS analyses, Variable Importance Projection (VIP) values > 1 indicated the important influence of one metabolite change in the BE, and this coefficient denoted the relationship (positive or negative) of one metabolite change with the BE. According to the result of OPLS, there were six amino acids’ VIPs > 1 in the leaves, among which five were positively correlated with the leaf length and leaf fresh weight while also appearing negatively correlated with the leaf MDA content (Figure 4A–C, Appendix A). In contrast, there were 12 amino acids’ VIPs > 1 in the roots, among which nine were negatively correlated with root length and fresh weight but positively correlated with MDA (Figure 4D–F, Appendix A). Under the Cd_10_ treatment, the soluble protein content in the leaves was 2.58 times that of the roots, and the higher soluble protein content had certain detoxification effects. In addition, to maintain the higher soluble protein content, the leaves consumed more amino acids, which resulted in a reduction in the content of important amino acids and affected the growth of the leaves, thus showing positive relationships with leaf length and fresh weight. In contrast, under the Cd_10_ treatment, the Cd content in the roots was 15.00 times that of the leaves; therefore, the roots were subjected to more toxicity than the leaves. The high Cd content could break down some proteins into amino acids while blocking the synthesis of amino acids into certain proteins, resulting in an elevated amino acid content in the roots [73,74]. The increase in the amino acid content was also needed for roots to resist high cadmium stress, which could lead to insufficient amino acid content for plant growth and ultimately inhibit plant growth, resulting in a decrease in the root length and weight with a negative correlation. The higher MDA content indicated severer oxidative stress caused by the Cd treatment; therefore, the increase in the amino acids could chelate the Cd ions and abate oxidative stress, showing positive relationships with the MDA content. In the leaves, there were two different carbohydrates’ VIPs > 1, which were positively correlated with the leaf length and leaf fresh weight but negatively correlated with MDA (Figure 4A–C, Appendix A). There were also two differential carbohydrates’ VIPs > 1 in the roots, which were positively correlated with the root length and fresh weight while also appearing negatively correlated with MDA (Figure 4D–F, Appendix A). Similar results were also found in organic acids such as citrate, malic acid, and pyruvaldehyde. It has been reported that heavy metal stress can destroy the chloroplast structure, reduce the ability of plants to capture light energy, and inhibit photosynthesis. This may be because the heavy metal cadmium replaced the plasma of iron and magnesium, which was bound with the sulfhydryl group on the chloroplast protein, reduced the activity of the chlorophyll enzyme, inhibited the synthesis of chlorophyll, and led to a decrease in the photosynthetic rate, thus decreasing the content of carbohydrates and organic acids and inhibiting plant growth [75]. Therefore, the changes in carbohydrates and organic acids showed positive relationships between the length and fresh weight and negative relationships with the MDA content. There were two differential nucleotide VIPs > 1 in the leaves, which were negatively correlated with leaf length and leaf fresh weight but positively correlated with MDA (Figure 4A–C, Appendix A). In comparison, three differential nucleotides’ VIPs > 1 in the roots, which were positively correlated with root length and fresh weight, were also negatively correlated with MDA (Figure 4D–F, Appendix A). Nucleotides are closely related to protein synthesis. The two nucleotides in the leaves were increased, while the three nucleotides in the roots were decreased, which partially confirms that Cd hindered protein synthesis in the roots.

### 2.6. The Metabolic Pathway Changes under Cd Stress

The metabolic pathways significantly changed under Cd stress. The Cd ions not only significantly impacted more metabolites and metabolic pathways in the roots than in the leaves but also indicated the larger impact of Cd stress on the roots than on the leaves. The metabolic pathways analysis showed that the Cd stress significantly altered seven pathways in the leaves, including glyoxylate and dicarboxylate metabolism (0.06, 2.03), purine metabolism (0.15, 1.83), pentose and glucuronate interconversions (0.1, 1.56), the citrate cycle (TCA cycle; 0.15, 1.42), amino sugar and nucleotide sugar metabolism (0.14, 1.39), zeatin biosynthesis (0.03, 1.39), and pyruvate metabolism (0.15, 1.35) (Figure 5A). In contrast, Cd stress significantly altered eight pathways in the roots, including the purine metabolism (0.16, 3.68), glycine, serine, and threonine metabolism (0.02, 3.48), arginine and proline metabolism (0.25, 2.25), aminoacyl-tRNA biosynthesis (0, 1.93), phenylalanine metabolism (0.19, 1.91), nicotinate and nicotinamide metabolism (0.21, 1.81), arginine biosynthesis (0.17, 1.42), and cysteine and methionine metabolism (0.09, 1.37) (Figure 5B).

Among the top 10 metabolism pathways in the roots or leaves, three shared metabolic pathways existed in both the roots and leaves, including purine metabolism, aminoacyl-tRNA biosynthesis, cysteine metabolism, and methionine metabolism. Purine metabolism is the only metabolic pathway that was shared in both the leaves and roots that was significantly affected by Cd stress, which was also closely correlated with nucleotide metabolism and was found to be significantly altered under Cd stress and other stresses (Figure 6). It has been reported that purine metabolism plays a pivotal role in the accumulation, tolerance, and detoxification of Cd in *Amaranthus hypochondriacus* ‘K472’. In addition to the purine metabolism, aminoacyl-tRNA biosynthesis was found to be significantly changed under Cd stress [6], salt stress [76], chilling stress [77], and drought stress [40]. The aminoacyl-tRNA biosynthesis was strongly associated with significant changes in the protein metabolism and the large accumulation of amino acids (Figure 6), leading to phenotypic damnification in fountain grass. According to a transcriptomic analysis, the cysteine and methionine metabolism pathways also significantly changed under Cd stress [78]. Plants could produce cysteine-rich peptides and chelate Cd ions to produce non-toxic complexes that are transferred into the vacuoles to obviate high levels of free cytotoxic Cd in the cytosol [79].

## 3. Materials and Methods

The seeds of fountain grass with full granules and uniform size were superficially sterilized by their contact with ethanol solutions (75% *v*/*v*) for 1 min and were washed with distilled water 5 times. The seeds were incubated in 0 (distilled water, control, CK), 5 mg/L (Cd_5_), and 10 mg/L (Cd_10_) Cd sulfate solutions in Petri dishes. Based on the pre-experiment for germination rate and fresh weight under each treatment, the numbers of seeds used for the experiment were 3000, 4000, and 11,000 for CK, Cd_5_, and Cd_10_, respectively. Every 50 seeds were cultured in a Petri dish with 9 cm diameter. Therefore, 60, 80, and 220 petri dishes were used for CK, Cd_5_, and Cd_10_, respectively. The Petri dishes were kept in incubators with 25 °C and 16 h light (6000 lux)/8 h dark photoperiod. Plant individuals or tissues were randomly selected for index determination on day 14 based on the seeds that germinated on day 3.

### 3.1. Plant Growth Parameters

On the 14th day, after imposing Cd stress, 30 seedlings were randomly selected as replicates, and 3 replicates were used for each treatment to measure the leaf/root length (270 seedlings in total). One hundred seedlings were randomly selected as replicates, and three replicates were used for each treatment to measure leaf/root fresh weight (900 seedlings in total).

### 3.2. Cd Contents

Each sample is 0.2 g of leaf or root tissue, and 3 samples were randomly selected for the leaves and roots of each treatment. Therefore, there are 9 leaf samples and 9 root samples in total. Each sample was dried and ground, followed by mixing with a 12 mL solution of concentrated nitric acid, hydrogen peroxide, and perchloric acid (4:1:1) in a PTFE beaker. The ground tissue in the beaker was fully digested on an electric heating plate at 180 °C. Then, the mixture was heated to near-dryness before adding 0.5% (*v*/*v*) nitric acid to a constant volume of 25 mL [4]. The Cd content (mg/g dry weight) was determined via inductively coupled plasma emission spectrometry (Optima 2100dv, Shelton, CT, USA). The transport factor was calculated as the ratio of Cd concentration in shoot to root [80].

### 3.3. MDA, Superoxide Dismutase Activity, and Soluble Protein Content Assay

For the determination of MDA content and superoxide dismutase activity, each sample is 0.5 g of leaf or root tissue, and 3 samples were randomly selected for the leaves and roots of each treatment. Therefore, there are 9 leaf samples and 9 root samples in total. Each sample was ground in 10 mL of a cold phosphate buffer (62.5 mM, pH 7.8), followed by 10 min (15,000 g, 4 °C) of centrifugation to obtain a supernatant. The supernatant was utilized to determine the superoxide dismutase (SOD) activity, peroxidase (POD) activity, catalase (CAT) activity, and MDA content. To determine the SOD activity, the reactive solution mixed up 0.2 mL enzyme solution, 2.1 mL phosphate buffers (62.5 mM, pH 7.8), 0.1 mL riboflavin (0.06 mM), 0.3 mL methionine (130 mM), 0.1 mL EDTA-Na_2_ (0.3 mM), and 0.2 mL NBT (1.125 mM). The SOD activity was determined by measuring the 560 nm absorbance [81]. To determine the POD activity, the reactive mixture contained 0.2 mL enzyme solution, 3 mL guaiacol (0.56 mL/L), and 0.8 mL 30% H_2_O_2_ (2 mL/L). The POD activity was determined by monitoring the increase in absorbance at 470 nm [81]. For the CAT activity assay, the reactive mixture consisted of 0.2 mL enzyme solution, 1.5 mL phosphate buffers (62.5 mM, pH 7.8), 1.0 mL deionized water, and 0.3 mL H_2_O_2_ (0.1 mM). The CAT activity was determined by observing the decrease in 240 nm absorbance [82]. To determine MDA content, the mixture comprised 1 mL of enzyme solution, 1 mL of phosphate buffers (50 mM, pH 7.0), and 2 mL of 10% trichloroacetic acid (containing 0.5% thiobarbituric acid). The mixture was centrifuged for 10 min at 4000× *g* after being water-bathed at 100 °C for 15 min. The absorbance values of the supernatant were measured at 450, 532, and 600 nm to calculate the MDA content [83].

For the determination of soluble protein content, each sample is 0.2 g of leaf or root tissue, and 3 samples were randomly selected for the leaves and roots of each treatment (9 leaf samples and 9 root samples in total). Each sample was ground in 5 mL of distilled water and then centrifuged (10,000 r/min, 10 min, 4 °C) to obtain the supernatant. The 1 mL of supernatant was fully mixed with 5 mL of Coomassie Brilliant Blue G250 solution and placed for 2 min. Then, the content of soluble protein was determined by measuring absorbance at 595 nm using bovine serum albumin as a standard [84].

### 3.4. Metabolite Detection and Analysis

On the 14th day, the randomly selected leaf and root tissues (0.2 g) in CK (3 leaf samples and 3 root samples) and Cd_10_ (3 leaf samples and 3 root samples) were inactivated with liquid nitrogen and ground in a 2 mL solution (methanol:water = 3:1). Then, each sample was homogenized and sonicated (in an ice-water bath for 5 min) by repeating 3-times, followed by 1 h at −40 °C for the incubation and 15 min (13,800 g, 4 °C) centrifugation to obtain the supernatant. The supernatant was used to determine the metabolites by LC–MS, which was carried out using an ultrahigh-performance liquid chromatography system (Thermo Fisher, Germering, Germany) coupled to a Q Exactive HFX mass spectrometer (Thermo Fisher, Bremen, Germany). The supernatant was injected into an Hyperil Gold column (100 × 2.1 mM, 1.9 μm) at a flow rate of 0.2 mL/min utilizing a 16-min linear gradient. There were 2 eluents for the positive polarity mode: eluent A (0.1% FA in water) and eluent B (methanol). There were also 2 eluents for the negative polarity mode: eluent A (5 mM ammonium acetate, pH 9.0) and eluent B (methanol). The settings of the solvent gradient were: 1.5 min 2% B; 12.0 min 2–100% B; 14.0 min 100% B; 14.1 min 100–2% B; 17 min 2% B. Q Exactive HFX mass spectrometer was employed in positive/negative polarity mode with a 3.2 kV spray voltage, a capillary temperature of 320 °C, a 35 arb sheath gas flow rate, and a 10 arb aux gas flow rate. The raw data were handled in Compound Discoverer 3.1. The procedures for each metabolite contained peak alignment, peak picking, and quantitation. The main criteria were retention time tolerance, 0.2 min; actual mass tolerance, 5 ppm; signal intensity tolerance, 30%; signal/noise ratio, 3; and minimum intensity, 100,000. The peak intensities were then normalized to the full spectral intensity. The normalized data was applied to predict the molecular formula by adding ions, molecular ion peaks, and fragment ions. Thereafter, accurate qualitative and relative quantitative results were obtained by matching the peaks with the mzCloud, mzVault, and MassList databases. The metabolites were annotated using the KEGG database, the HMDB database, and the Lipidmaps database [85]. The metabolite with a fold change >2.0 or <0.5 and *p <* 0.05 compared with the control was discriminated against as a differential metabolite.

### 3.5. Statistical Analysis

R v. 4.2.0 was used for one-way ANOVA (*p* < 0.05) to evaluate the effects of cadmium stress on morphological and physiological characteristics [86]. The post hoc test with the least significant difference proceeded to compare the morphological and physiological differences among each level of cadmium treatment. The heatmap of differential metabolite changes was conducted by the Statistical Analysis/Cluster Analysis/Heatmaps module in MetaboAnalyst. SIMCA 14.1 (Sartorius, Gottingen, Germany) was employed for the orthogonal partial least squares regression analysis (OPLS) to explore the relationships between differential metabolite changes and the BEs. Then, the relationships of differential metabolite changes with BEs were visualized in Cytoscape 3.9.1 [87].

## 4. Conclusions

Cd stress restrained plant growth and reduced leaf length, leaf fresh weight, root length, root fresh weight, and total weight; however, it also increased the MDA content and soluble protein content. Under Cd stress, most of the differential amino acids and differential lipids decreased in the leaves while increasing in the roots. Most of the carbohydrates in leaves and roots decreased under Cd stress, which contributed to a decrease in leaf leaf/root length and fresh weight. Almost all the differential amino acids in roots were negatively correlated with root length and root fresh weight, while they were also positively correlated with MDA content. Metabolic pathway analysis showed that Cd significantly affected seven and eight metabolic pathways in the leaves and roots, respectively, with only purine metabolism shared in both the roots and leaves. Fountain grass adapts to Cd stress by adjusting the above metabolic pathways in leaves and roots. Metabolites that largely increased under Cd stress in our study could be exogenously applied to other plants to test their mitigation effects in various adverse environments. Transcriptomics and proteomics could be applied to validate the purine metabolism to better understand the mechanism by which fountain grass copes with Cd stress in the future.

## Figures and Tables

**Figure 1 plants-12-03713-f001:**
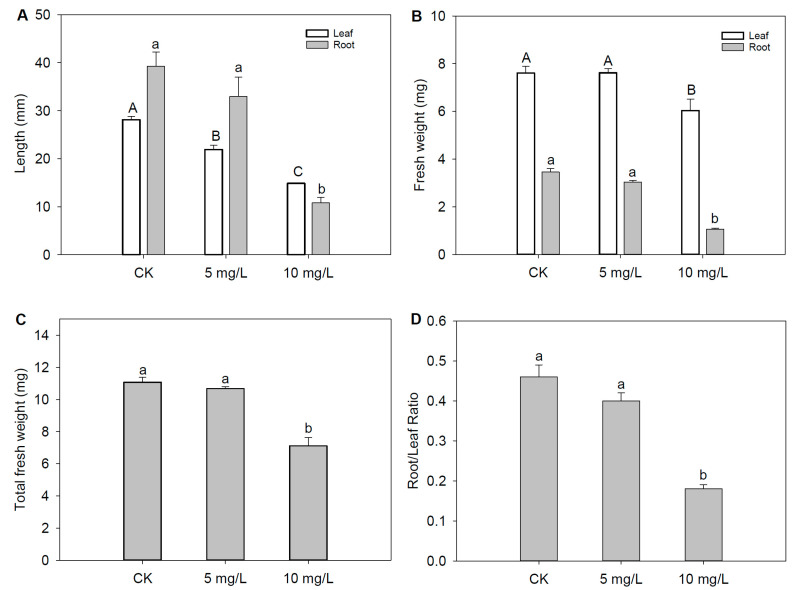
Growth characteristics of fountain grass under cadmium (Cd) stress. (**A**) Length of leaf and root. (**B**) Fresh weight of leaf and root. (**C**) Total fresh weight. (**D**) Root/Leaf ratio. Different characters denote significant differences at *p* < 0.05 (*n* = 3). Vertical bars indicate standard error.

**Figure 2 plants-12-03713-f002:**
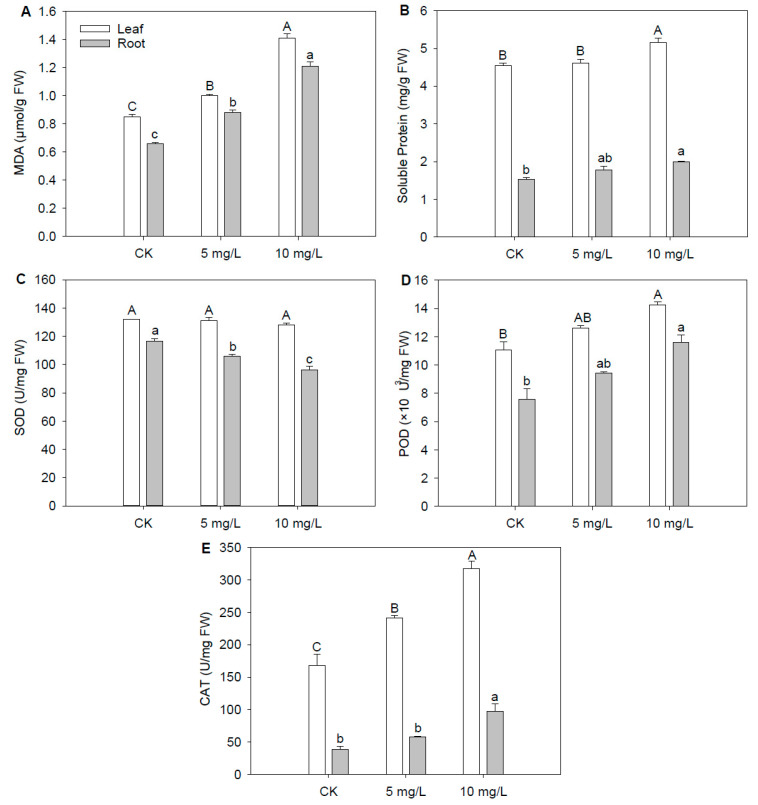
The MDA content, soluble protein content, and SOD, POD, and CAT activities of fountain grass under Cd stress. (**A**) MDA. (**B**) Soluble protein. (**C**) SOD. (**D**) POD. (**E**) CAT. Different characters denote significant differences at *p* < 0.05 (*n* = 3). Vertical bars indicate standard error.

**Figure 3 plants-12-03713-f003:**
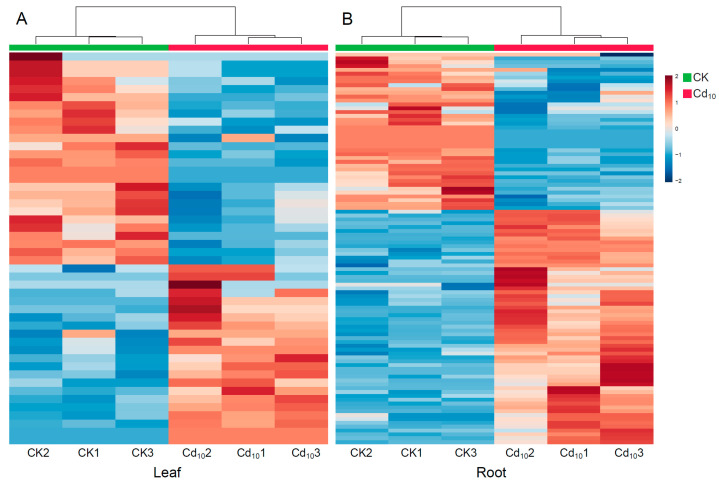
Heatmap clustering of the relative contents of the differential metabolites of fountain grass under Cd stress (*n* = 3). Each row represents a metabolite, and each column represents a sample. The warmer color indicates a higher content, and the colder color denotes lower content. (**A**) Leaf. (**B**) Root.

**Figure 4 plants-12-03713-f004:**
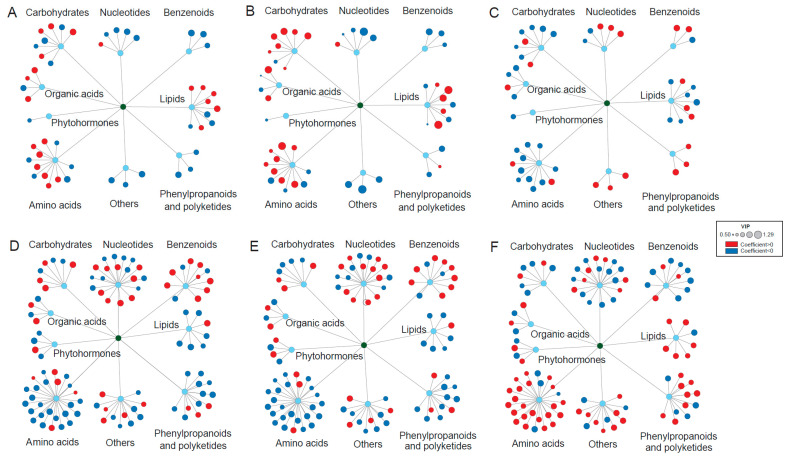
Relationships between the differential metabolite changes and biological endpoints (BEs) of fountain grass induced by Cd stress (*n* = 3). (**A**) Leaf length as the BE. (**B**) Root length as the BE. (**C**) Leaf fresh weight as the BE. (**D**) Root fresh weight as the BE. (**E**) Leaf MDA content as the BE. (**F**) Root MDA content as the BE. The red and blue circles represent active and passive coefficients, respectively. The sizes of the circles represent the variable importance projection (VIP) values.

**Figure 5 plants-12-03713-f005:**
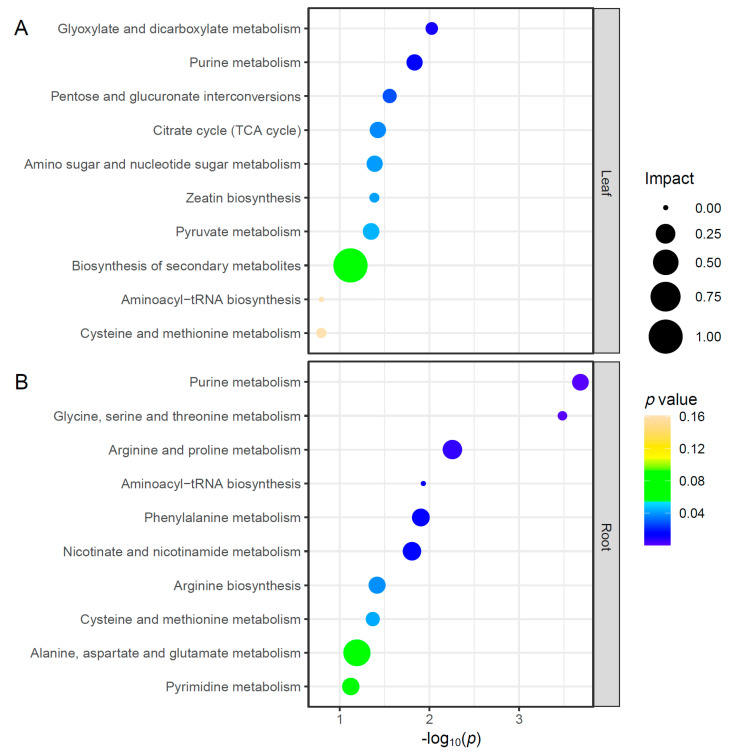
Top 10 metabolic pathways in the leaf or root of fountain grass affected by cadmium stress via pathway analysis from differential metabolites (*n* = 3). The pathway impact was computed from the pathway topological analysis and represented via the circle size. The *p* values obtained from pathway enrichment analysis were shown both by the *x*-axis (−log_10_) and color. (**A**) Leaf. (**B**) Root.

**Figure 6 plants-12-03713-f006:**
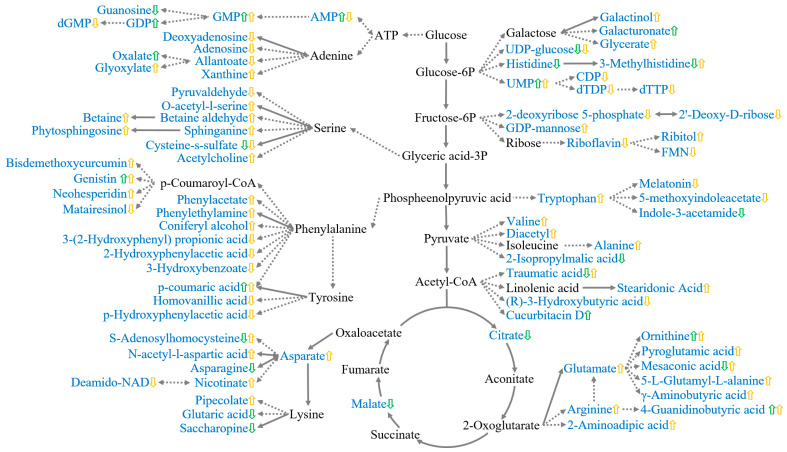
Effects of cadmium stress on the metabolic map of fountain grass (*n* = 3). The metabolites in blue text were the differential ones. Green or yellow arrows denote metabolites in the leaf or root, respectively. The up or down arrow direction indicates if the metabolite was up-regulated or down-regulated under cadmium stress. The linking arrows among metabolites with full or dotted lines denote the direct or indirect reactions, respectively. (For an interpretation of these references regarding color in this figure legend, please refer to the web version of this article.)

**Table 1 plants-12-03713-t001:** Cadmium (Cd) contents in the leaves and roots of fountain grass seedlings under cadmium stress.

Treatment	Cd Content in Roots (mg/g Dry Weight)	Cd Content in Leaves (mg/g Dry Weight)	Transfer Factor
CK	0.00 (C, a)	0.00 (C, a)	/
Cd_5_	0.50 ± 0.06 (B, a)	0.04 ± 0.00 (B, b)	0.08
Cd_10_	1.05 ± 0.21 (A, a)	0.07 ± 0.01 (A, b)	0.07

Different letters indicate significant differences at *p <* 0.05 (*n* = 3). Uppercase letters indicate significant differences in the Cd content among different treatments in the same tissue; lowercase letters indicate significant differences in the Cd content between roots and leaves of the same treatment.

## Data Availability

Not applicable.

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
