# Peer review of "The Influence of Cadmium on Fountain Grass Performance Correlates Closely with Metabolite Profiles"

_plants, 2023, doi:10.3390/plants12213713_

Round 1
Reviewer 1 Report
Comments and Suggestions for Authors
In this manuscript, the authors have analyzed the morphological and the physiological responses of fountain grass to different Cd stress and analyzed the metabolic responses by detecting the metabolites through UPLC-MS. The results showed that Cd restrained plant growth, reducing leaf length, leaf fresh weight, root length, root fresh weight, and total weight; however, it also increased the malondialdehyde content and soluble protein content. In summary, under Cd stress, 102 differential metabolites in roots and 48 differential metabolites in leaves were detected, with 20 shared metabolites. Under Cd stress, most of the differential amino acids and lipids decreased in the leaves but increased in the roots. Most of the carbohydrates in leaves and roots decreased under Cd stress, which contributed to the lowered leaf/root length and fresh weight. Almost all the differential amino acids in the roots were negatively correlated with the root length and root fresh weight, while they were positively correlated with malondialdehyde content.
The paper is well designed and written. However, the English grammar needs revisions.
The abstract should include a clear objective too.
The methods are described sufficiently.
The results are well representative.
The discussion should include an improved comparison with the present literature.
The conclusions should include a future recommended work based on the current results as well.
Comments on the Quality of English Language
Minor editing of English language required
Reviewer 2 Report
Comments and Suggestions for Authors
I have had the pleasure of reviewing your manuscript titled " Cadmium-Induced Fountain Grass Performance Is Highly Related to Metabolite Pattern." I must commend your comprehensive and robust approach to this pertinent issue in the realm of agricultural science.
The issue of Cadmium stress is indeed a pressing one, and your research highlights the problem and provides a tangible and innovative discussion. However, there are some crucial amendments required as follow:
Point 1: The suggested title of the MS should accurately represent the performed experiments without unnecessary information. I would suggest the title to be “The Influence of Cadmium on Fountain Grass Performance Correlates Closely with Metabolite Profiles”
Point 2: In the Abstract section, in lines 21-22, the authors noted the detection of "102 differential metabolites in roots and 48 differential metabolites in leaves under Cd stress, with 20 shared metabolites." However, I find it challenging to discern how this conclusion was arrived at based on the results and discussion provided in the manuscript. I attempted to deduce this conclusion from the presented data but was unable to arrive at a satisfying explanation. It appears that there may be some crucial details missing from the results that are needed to support this assertion.
Point 3: In the Methods and Materials (M&M) section, it is important for the authors to provide a clear description of the methodology used to evaluate Differential Amino, Differential Carbohydrates, Differential Lipids, Differential Organic Acid, and Differential Phytohormone, as these measurements were mentioned in both the results and discussion sections. Therefore, it is crucial that the methodology for assessing these variables is explained in detail to bridge this gap in the research presentation.
Point 4: In the results and discussion section, there is a noticeable absence of specific results pertaining to Differential Amino, Differential Carbohydrates, Differential Lipids, Differential Organic Acid, and Differential Phytohormone measurements in the manuscript.
Point 5: In addition to the point I've raised previously I noticed that further information could be beneficial in the captions of your tables and figures, particularly regarding the number of replicates used in your experiments.
Including information about the number of replicates (n) in your captions is important for several reasons. Firstly, it provides the reader with a clear understanding of the sample size, which can influence the interpretation of the data, especially when considering variability and statistical significance.
Point 6: While your manuscript displays a strong command of the topic and presents compelling findings, I noticed some minor language issues and inconsistencies throughout the text. These could potentially hinder the clarity of your message and disrupt the reader's engagement with your work.
Comments on the Quality of English LanguageI noticed some minor language issues and inconsistencies throughout the text. These could potentially hinder the clarity of your message and disrupt the reader's engagement with your work.
Reviewer 3 Report
Comments and Suggestions for Authors
Cadmium-Induced Fountain Grass Performance Is Highly Related to Metabolite Pattern
Abstract – is extensive. Only the most important information should be presented in this section. Please rewrite it.
Introduction - There is a lack of information why the authors chose fountain grass to their research. Does the plant play a significant role in agronomy, agriculture or the environment? Is it a model research plant? Please explain
Materials and methods – How was Cd concentration chosen for research? Why authors tested 5 and 10 mg/L Cd concentration?
What is the vegetation period of Fountain Grass? At what development stage was plants taken for analyzes? According to the authors, whether conducting research on plants of the same species but at a different development stage would allow you to obtain the same results regarding no. metabolites?
Line 445 – what was the detection limit for Cd analysis?
Lines 457 and 486-487 – Sentences are not consistent. Please correct
Lines 486-487 – why plant samples treated by Cd in concentration 5 mg/L were not tested? Please explain
Results and discussion
Lines 158-165 - Paragraph is too speculative. The authors did not provide such research, therefore these sentences should be summarized by the statement that only quantitatively soluble proteins were marked, without analyzing what kind of protein that were synthesized by the plant under Cd stress conditions. Further research in this area is necessary.
Figure 2 - How can you explain the higher content of soluble proteins in the leaves of the plant compared to the roots?
Figure 3 – unreadable heat map. Please improve it.
Lines 104, 182 - Please provide the year of cited work in brackets
Round 2
Reviewer 2 Report
Comments and Suggestions for Authors
I am pleased to note that the authors have made substantial improvements to their work based on my initial feedback.